# Estimation of the Lateral Distance between Vehicle and Lanes Using Convolutional Neural Network and Vehicle Dynamics

**Xiang Zhang, Wei Yang *, Xiaolin Tang * and Zhonghua He**

State Key Laboratory of Mechanical Transmission, College of Mechanical Engineering, Chongqing University, Chongqing 400044, China; zkebi@126.com (X.Z.); bushuohua704@126.com (Z.H.)

\* Correspondence: slmt053@cqu.edu.cn (W.Y.); tangxl0923@cqu.edu.cn (X.T.);
Tel.: +86-359-469-5699 (W.Y.); +86-511-199-0183 (X.T.)



**Featured Application:** **In this study, two cameras were mounted at the side mirrors of the vehicle and a lateral lane distance detection algorithm was designed to accurately estimate the lateral distance between the vehicle and the lane. The research results are applicable to the applications of high lateral position accuracy of the vehicles, for instance, testing and evaluating the performance of some applications in ADASs (Advanced Driver Assistance Systems), such as LKAS (Lane Keeping Assist Systems), LDW (Lane Departure Warning), etc. Moreover, it can also be used in the design of applications such as LDW, LKAS and Parking Assist, etc.**

**Abstract:** With the aim to achieve an accurate lateral distance between vehicle and lane boundaries during the road test of Lane Departure Warning and Lane Keeping Assist, this study proposes a recognition model to estimate the distance directly by training a deep neural network, called LatDisLanes. The neural network model obtains the distance using two down-face cameras without data pre-processing and post-processing. Nevertheless, the accuracy of recognition is disrupted by inclination angle, but the bias is decreased using a proposed dynamic correction model. Furthermore, as training a model requires a large number of label images, an image synthesis algorithm that is based on the Image Quilting is proposed. The experiment on test data set shows that the accuracy of LatDisLanes is 94.78% and 99.94%, respectively, if the allowable error is 0.46 cm and 2.3 cm when the vehicle runs smoothly. In addition, a bigger error can be caused when inclination angle is greater than 3°, but the error can be reduced by proposing a dynamic correction model.

**Keywords:** lateral distance estimation; image synthesis; inclination angle; convolution neural network; vehicle dynamics

## 1. Introduction

Thanks to the development of big data and computer vision and other advanced technology [1–3], modern cars are equipped with a large number of Advanced Driver Assistant Systems (ADASs), such as Lane Departure Warning (LDW) and Lane Keeping Assist Systems (LKAS), which are designed to perform the functions of lane departure warning and keeping [4]. However, before vehicles with these safety applications are launched on the market, numerous road tests ought to be carried out. Suggestions and basis for improvement are provided by evaluating the performance of LDW and LKAS systems [5,6].

The performances of LDW and LKAS are related to the accurate positioning of a vehicle between two lane boundaries and the required accuracy is almost 100% [7]. The lane recognition based on machine vision is mainly composed of feature extraction and lane detection [8]. The validity

of extracted features determines the accuracy of lane detection. The contour information of lane is generally extracted, then the contour information is matched by the modelling of lane, and the location of lane in the image is obtained. For example, Canny, Robert, Prewitt, and Sobel are used to find shapes in images, Gaussian filters are used to smooth images after edge detection with Canny operators, and then lanes are fitted by the Hough Transform or Random Sample Consensus (RANSAC) algorithm [9–11]. However, these edge detection algorithms not only obtain lane contour model, but also bring a lot of other noises, such as the contour of roadside buildings and driving vehicles, etc., which brings inconvenience to the subsequent denoising.

Most lanes appear in the collected images in the form of straight lines, and the Hough Transform is a standard edge extractor that has significant advantages in finding lines in the image, so it is often used for lane detection. Collado et al. [12] broke the problem down into several steps: create a bird-view of the road, divide pixels belonging to longitudinal road markings, use the Hough Transform to extract lane boundaries, and adjust pitch angles. In [13], the parameter equation of the line to be matched is obtained by using the Hough Transform and the lane is extracted by RANSAC verification in the straight line constraint box, which can meet the real-time requirements. The lane detection method that is based on the Hough Transform has good robustness to the straight lane, but the robustness is reduced on curved and damaged lanes.

In order to enhance the feature representation of the lane, the colour information is also fully utilized. The colour information based lane detection algorithm usually converts the colour mode RGB (Red, Green, and Blue) to HSI (Hue, Saturation, and Intensity), which can greatly reduce the colour composition of the road surface when the shadow or light is insufficient [14]. In [15], the lane marking features are extracted by analysing the colour distribution of road, but the performance of the extractor will be affected by drastic illumination changes. Borkar et al. [16] used adaptive local thresholds to generate a binary edge image with good performance at edges. Yoo et al. [17] used gradient enhancement transformation to produce gray-scale images with large gradients at lane boundaries. Du et al. [18] designed a filter to extract lane markings from grayscale images. The disadvantage of these algorithms is that they are seriously affected by the brightness change of light. Additionally, other vehicles on the road often interfere with lane identification.

Stereo vision [19] technology can be used to distinguish lanes and obstacles, but stereo vision requires high accuracy of camera calibration parameters and the algorithms are relatively complex. In order to reduce the effect of sensors on the detection accuracy, some scholars have combined the information of GPS, high-precision map, and LiDAR (Light Detection And Ranging) [20] to detect the lane of road surface. However, GPS is greatly affected by satellite signal intensity, and the cost of high-precision maps and LiDAR is high, which limits the large-scale application.

Luckily, deep learning technologies have been developed rapidly in recent years and have been applied to the lane detection and tracking field as an extension of the traditional machine learning [21]. Li [22] raised a lane recognition algorithm that sufficiently extracts features of road surface by convolution neural network (CNN) and recurrent neural network (RNN). Kim [23] presented an algorithm for image improvement based on CNN, and completed lane detection with the RANSAC. Brust [24] proposed a traffic scene segmentation algorithm on the basis of CNN to enhance the accuracy of lane detection. Although satisfactory results in lane detection can be achieved, post-process is still needed to locate the position of vehicle between two lane boundaries.

Therefore, this study proposes a CNN-based recognition model that can obtain the lateral distance between vehicle and lane strip directly from the input image without pre-processing and post-processing. However, the training of the high-accuracy CNN model usually requires a lot of label images, and building a ground-truth database is complex and time-consuming [25]. Currently, the commonly used method is to mark the lane position in every frame of a video, and the processing of each frame takes about one minute, that is to say, it would take about one month to label a one-hour video [26]. Hence, a more advanced algorithm with higher efficiency of label images generation is needed. Borkar [27] proposed a ground truth generation algorithm that is based on time-slice, wherein

every frame needs only two labelled points and the ground truth can be obtained by stacking the frames from the bottom to the top. In order to speed up the labelling process, Al-Sarraf [28] proposed an improved time-slice algorithm, where lane pixels were located quickly with the help of Canny. With this approach, the efficiency of manual labelling was improved greatly, but it still depends on manual annotation.

Since the generation of the label data requires a large number of images and manual annotation, which is cumbersome and time-consuming, and accuracy of the label is greatly affected by human factors, some alternative algorithms for label image generation were studied and some trained model has achieved promising results. Revilloud [29] developed a lane detection algorithm to achieve a centimeter-level recognition accuracy using 2500 synthetic images on the SIVIC platform. Gurghian [30] trained a deep learning model for lane detection using 80,000 real scene images and 40,000 synthetic images, and the accuracy of detection reached 94%. Using these image synthetic algorithms, the label images can be obtained on demand, and for this reason, the efficiency of model training can be improved greatly. Nevertheless, the speed of image synthesis is usually unsatisfactory. Meanwhile, the condition of light, road surface wear and pollution on the road are not taken into consideration. Thus, a new algorithm for synthesizing label images with factors of real traffic scene is needed.

Among the many existing image synthesis algorithms, Image Quilting [31] is the most commonly used. The texture information on the original image can be well preserved. Due to the obvious advantage and spotlight of Image Quilting, the algorithm has attracted much attention from researchers. Kwatra [32] proposed an improved Image Quilting algorithm that is based on the Graph Cut and solved the problem of boundary distortion on the basis of graph theory. Dong [33] introduced the Genetic Algorithm to search the nearest neighbor in an adaptive way to match the texture block, with the efficiency being significantly improved. Long [34], by taking the Partial Non-Scalar Distance as a minimum path metric, eliminated the seam, and obtained the synthetic images with high quality. However, the RGB information on the image was used as the only matching metric in the above methods, there were still distortion and warping in the boundary, especially in the images that accompany with stronger structural information [35]. Moreover, selecting a new texture block in a random way for image synthesis is time-consuming.

In this study, an image synthesis algorithm was proposed, which only needs to collect a small number of images in a real scene, and can synthesize a large number of images as needed. Subsequently, a label automatic annotation algorithm is presented to provide the label images for training the CNN model that is designed to detect the lateral distance between the vehicle and lane strip. Due to external load, suspension, and tire during high-speed driving, a large inclination angle would be produced. For this reason, a big recognition error of LatDisLanes may be caused. Therefore, this study proposes a dynamic correction model to eliminate the effect of inclination angle on recognition accuracy. The main contributions of this work are as follows.

1.　An algorithm for image synthesis is proposed using only a few images from the real scene and a large number of high-quality images can be obtained.
2.　A label automatic annotation algorithm that provides the label images for training the CNN model based on the synthetic images is presented.
3.　A CNN model, LatDisLanes, is designed to recognize the lateral distance. Furthermore, to reduce the influence of inclination angle on model recognition accuracy, this study proposes a dynamic correction model.

This manuscript is structured as follows: In Section 2, a recognition model that is based on CNN is proposed. In order to train the model, an approach for generating label images is presented and the proposed dynamic correction model aiming to reduce the recognition error that is caused by inclination angle is proposed. The evaluation of proposed approach is given in Section 3 and concluding remarks are presented in Section 4.

## 2. Method

The main research methods are as Figure 1. First, an improved Image Quilting algorithm is proposed to provide label data for the training of a deep neural model. The images of lane as well as asphalt can be synthesized rapidly using few real-scene images. A label automatic annotation algorithm is also presented to provide the required label images for model training. Subsequently, a CNN model is constructed to estimate the lateral distance between the lane strip and the vehicle. In order to eliminate the error that is caused by the inclination angle, a dynamic correction model is proposed to correct the CNN recognition result, and the lateral distance is output accurately.

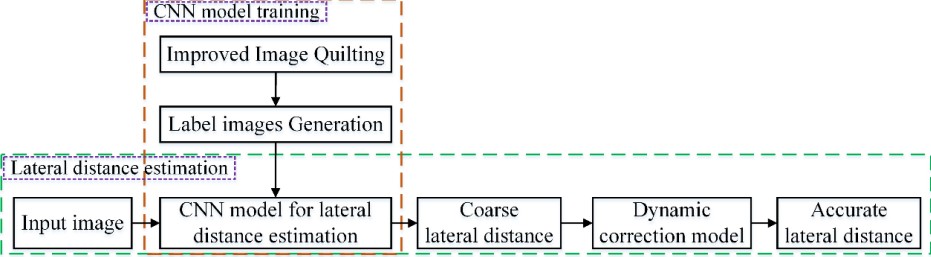

**Figure 1.** The flowchart of the algorithms. The "convolution neural network (CNN) model training" included label images generation method and model training. The "Lateral distance estimation" detects the lateral distance in the input image by using the trained model, and the bigger error was reduced through the dynamic correction model.

### 2.1. Method of Label Images Synthesis

The training of CNN model requires a large number of label images. Nevertheless, the procedure of images collection and annotation is complex and time-consuming. In fact, the asphalt pavement is composed of many small gravels that exist in a random distribution. We only need to cut the pavement and lane images in the real scene into small pieces, and then randomly combine them into new asphalt pavement and lane images with desired shape using an improved Image Quilting algorithm. In this way, only a small number of images in a real scene are needed to generate a large number of road and lane images. Furthermore, the label in the label image needed is the lateral distance between the vehicle and the lane strip. Usually, the lateral distance is not long, so a high-precision label needs to be determined. In this study, the label image is synthesized according to the pixel distance between the edge of lane and asphalt image, so that the accuracy of the label reaches the pixel level, and it is not affected by the subjective factors when the image is manually labelled. The generation process of the label image is shown in Figure 2.

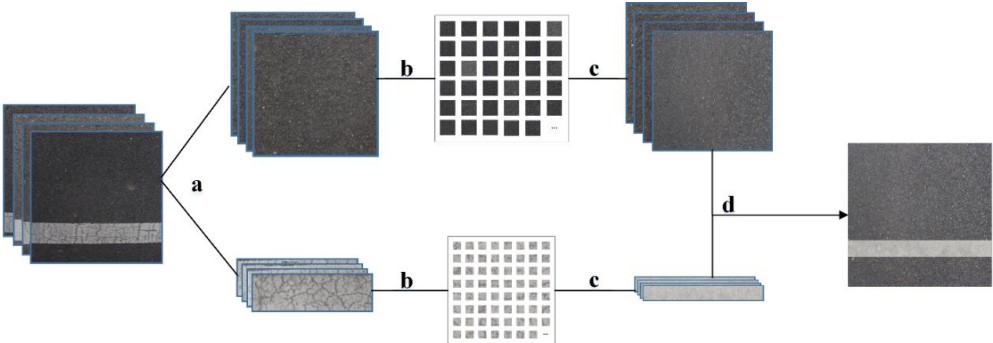

**Figure 2.** The framework of label images generation. Firstly, the lane and asphalt pavement in the original road image are separated manually (**a**), then the two type images are cut into small fragments (**b**). Then a large number of road and lane images are synthesized by the proposed improved Image Quilting method (**c**). Finally, the label images are synthesized according to the pixel distance between the lane strip and the asphalt pavement image (**d**).

2.1.1. Improved Image Quilting

In order to ensure the efficiency and quality of the generated images, this study proposes a fast and efficient image synthesis method that is based on the Image Quitting algorithm. The speed of image synthesis is greatly improved with guaranteed quality of images using the gradient and color information of the edges in the image obtained via the database technology.

In the Image Quilting, the image is synthesized by calculating the total error of overlap of the blocks to be synthesized and the synthesized one, and the color error is introduced as similarity metric.

$$cost = \sum_{\substack{p \in N_1, q \in N_2 \\ N_1, N_1 \in overlap}} \sqrt{(R(p) - R(q))^2 + (G(p) - G(q))^2 + (B(p) - B(q))^2} \tag{1}$$

where the *cost* is the error of two texture blocks in the overlap area *N*, and *R*, *G*, and *B* are the values of three primary colors, respectively. Obviously, the similarity is higher if the error of two texture blocks is smaller.

Since the asphalt is made up of bitumen and small gravels whose structural feature is more prominent, the synthesized images of asphalt are readily distorted and mismatched, thus taking the color error as the only similarity metric is obviously unreasonable. Moreover, each new texture block to be synthesized is selected randomly and the similarity is calculated repeatedly. Only if the similarity meets the requirement of similarity metric, the new block could be stitched to the synthesized area. Obviously, the image synthesis is time-consuming, and as a result the synthesis time is disturbed by the similarity metric. A small metric would prolong the image synthesis. Therefore, an improved algorithm for image synthesis that is based on Image Quilting is proposed and the improvements are as follows:

1.  Improvement of similarity matching metric

The gradient that represents the structure is added to the RGB and feature information on the texture block is defined by:

$$info(N) = \sum_{p \in N} [\alpha[R(p) + G(p) + B(p)] + \beta grad(p)] \tag{2}$$

where *p* denotes the pixel and grad is the gradient of the image. Furthermore, $\alpha$ and $\beta$ represent the contribution coefficients of color and gradient, respectively, and they satisfy condition $\alpha + \beta = 1$. The definitions of $\alpha$ and $\beta$ are:

$$\alpha = \frac{\sum_{p \in N} [R(p) + G(p) + B(p)]}{\sum_{p \in N} [R(p) + G(p) + B(p) + grad(p)]} \tag{3}$$

$$\beta = \frac{\sum_{p \in N} grad(p)}{\sum_{p \in N} [R(p) + G(p) + B(p) + grad(p)]} \tag{4}$$

The cumulative error of the overlap area is defined by:

$$cost = |info(N_1) - info(N_2)| \tag{5}$$

2.  Improvement of texture block selection

Since selecting a new texture block in a random way for image synthesis is time-consuming in the Image Quilting algorithm. An alternative method is introduced to accelerate the selection of texture

blocks. Before the image is synthesized, feature information on the edge of texture blocks is calculated by (2) and is stored into the database. Smaller cost means higher similarity. However, if the size of overlap area and blocks' type change, the value of cost may change as well. For improving the versatility, a similarity of texture blocks is defined by:

$$\lambda = \left(1 - \frac{|info(A') - info(B')|}{info(A')}\right) \times 100\% \tag{6}$$

The $\lambda_0$ is considered as a minimum of similarity metric and each similarity in the overlap should meet the condition $\lambda \geq \lambda_0$. Thus, the required feature information of the corresponding border at the required block is limited by:

$$\lambda_o info(A') \leq info(B') \leq (2 - \lambda_o)info(A') \tag{7}$$

During the process of image synthesis, the first block is selected in a random way and is placed on the top left of the area to be synthesized. Afterwards, the new block whose left feature information matches with the right feature of synthetic area according to the similarity metric is selected by (7) and then placed on the right of the synthesized area in Figure 3a. Similarly, the block whose top feature matches with the under feature of synthetic area is selected and placed under the synthesized area in Figure 3b. In the interior of synthesized area in Figure 3c, the required block whose left and top feature information should meet the requirement at the same time. In addition, the selected blocks will be stitched to the synthesized area by Dijkstra [36]. The required block in the corresponding area can be searched from the database using very little time. Accordingly, for a given size of the image, the synthesis time is constant.

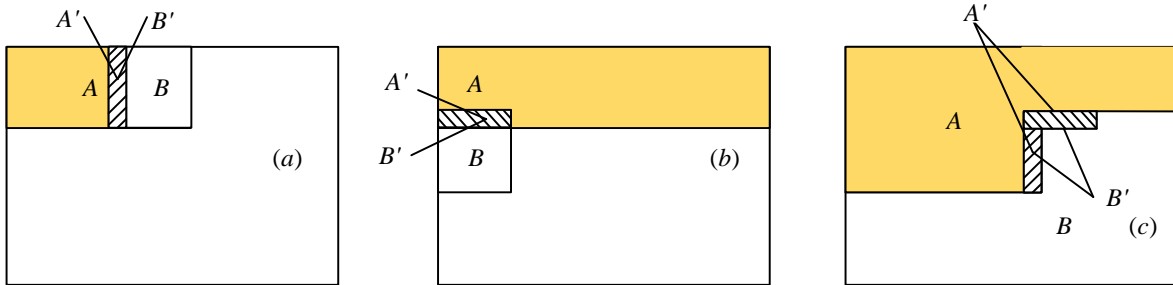

**Figure 3.** The hatched section shows overlap area. The (**a**) is the synthesis process of the first row. The (**b**) is the synthesis process of the first column, and the (**c**) is in both horizontal and vertical directions in the interior. *A* is the synthesized area and *B* is the block to be synthesized, and *A'* and *B'* are the overlaps of block A and B, respectively.

2.1.2. Generation of Raw Labelled Images

1.　Generation of road shape

In fact, the vehicle driving cycle is complex and varied, and there are diverse types of roads. Moreover, the lane is also marked as dotted, solid, or dotted-solid. Nevertheless, the label image generation method in this study has strong versatility. The lane and asphalt can be synthesized by the improved Image Quilting as desired shapes according to the specific traffic scene and regulatory standards.

2.　Lane edge filtering

The lane images that are synthesized by the improved Image Quilting are accompanied with a regular edge, whereas the images in real scene are usually serrated. We set the width of serrated area

on the edge of lane as $N_l$ wherein a nonlinear filter is adopted to replace some pixels of lane with the corresponding pixels in asphalt [37].

$$p(x,y) = (1 - \varepsilon) \cdot lane(x,y) + \varepsilon \cdot road(x,y) \tag{8}$$

where $\varepsilon$ is a random integer between 0 and 1.

3.   Uniform wear

The uniform wear of the road comes in two forms: disappearance of the lane mark and asphalt penetration. The former one appears generally on the top of gravels, whereas the latter is caused by the penetrating of bitumen, which can be considered as monochrome. Therefore, the pixels of lane are replaced by the ones of asphalt when the pixel value is lower than $L_{th}$ or higher than $H_{th}$.

$$road(x,y) \leftarrow \begin{cases} lane(x,y) < L_{th} \\ lane(x,y) > H_{th} \end{cases} \tag{9}$$

4.   Texture consistency

Lane usually has the same texture trend as asphalt due to wear and penetration of bitumen. In order to keep the texture of lane consistent with asphalt, the pixel of the lane is fixed by:

$$p(x,y) = lane(x,y) - \gamma \cdot (\mu + \delta - road(x,y)) \tag{10}$$

where $lane(x, y)$ and $road(x, y)$ are the pixel values of lane and asphalt, respectively. $(x, y)$ is the coordinates of asphalt, $\gamma$ is the asphalt influence coefficient whose value is between 0 and 1, $\mu$ and $\delta$ are the mean and variance of asphalt in a local square area with a length of $L_s$.

Finally, the asphalt is covered with a lane whose location is regulated by $d_i$. Meanwhile, the processes of lane edge filtering, uniform wear, and texture consistency are underway during the process of raw label images generation the obtained image is as shown in Figure 4.

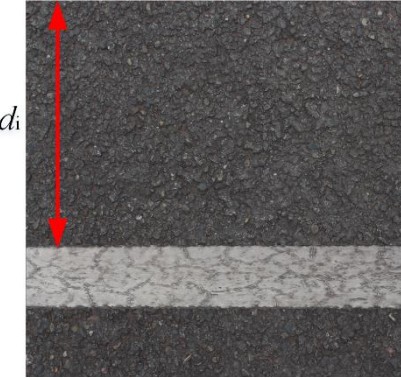

**Figure 4.** Raw label image. The $d_i$ is the distance from bottom edge of lane to asphalt.

2.1.3. Label Image Process

1.   Dirt addition

The synthetic raw image is clean and dirt-free, whereas normal roads inevitably expose to all kinds of dirt. One of the very common dirt is the tire traces caused by brake and other dirt, such as dust. In this study, only the tire trace and dust are taken into consideration. Tire trace is a structural dirt and it cannot be added at will, as the area and size of tire trace are adjusted and then covered to the raw label images. In addition, as the dust is usually randomly distributed. we adopted the model

presented in [38] to generate a mask with a random shape to cover on raw label images, and filled the mask area with dust pixels.

2.　　Shadow addition

The shadows in the real scene are generally induced by a shelter of buildings, vehicles, trees etc. On the one hand, as the shadows that are produced by buildings and vehicles are generally large, a constant can be subtracted from the pixel value in images. On the other hand, the shadow of trees is accompanied by a random shape, thus the mask generated in [33] is also adopted and the effect of light is achieved by increasing or decreasing the constant *c* within the mask:

$$road(x, y) \leftarrow road(x, y) - c \tag{11}$$

*2.2. Recognition Model for Estimating Lateral Distance*

The camera layout is shown in Figure 5, wherein it can be seen that two downward cameras are placed in the rear-view mirrors. Relying on this placement, the region of interest (ROI) is limited to a small area around the vehicle, and most of the disordered information in the images can be eliminated to obtain a satisfactory low-resolution camera [39].

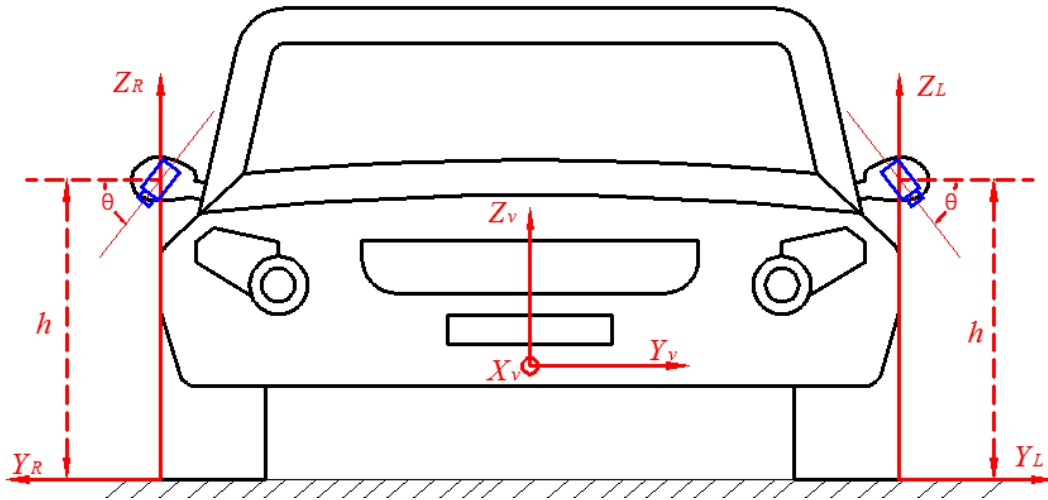

**Figure 5.** Extrinsic configuration of the side cameras and associated coordinate systems.

This study aims to build a model that can recognize a lateral distance between vehicle and lane boundaries. The $Y_L$ and $Y_R$ in Figure 5 denote the distance to the lane marking on left and right.

As CNN is widely used in image recognition due to its good robustness, high accuracy, and efficiency, inspired by the method in [30], a deep CNN is used in this study to acquire a lateral distance between vehicle and two-lane boundaries directly without pre-processing and post-processing.

The recognition of lateral distance is considered as a multitasking detection problem. For a given image $X_i$, the label that indicates the distance between lane strip and vehicle is $d_i$. The output vector of softmax layer is $Y_i = (y_0, y_1, \ldots, y_{320})$. In fact, there are possibly no lanes on the road surface, so we assumed $d_i = 321$. Furthermore, the entry $y_k$ is the probability that row $k$ in image $X_i$ contains the position of the lane marking.

After the image $X_i$ is fed through the network, the position of lane $e_i$ is assumed the image row in line with the entry in $Y_i$ with the maximum likelihood:

$$e_i = \underset{0 \leq k \leq 321}{\arg \max} \, y_k \tag{12}$$

In the training process of the model, the cross-entropy [40] that defined by (13) is introduced as the cost function, which is used as a criterion for measurement of the distinction between $e_i$ and corresponding label $d_i$.

$$J(\eta) = -\frac{1}{m}\sum_{i=1}^{m} y^{(i)} \log\left(h_\eta\left(x^{(i)}\right)\right) + \left(1 - y^{(i)}\right) \log\left(1 - h_\eta\left(x^{(i)}\right)\right) \tag{13}$$

where $h_\eta(x^{(i)}) = 1/(1 + e^{-\eta^T x^{(i)}})$ is the activation function, and $m$ is the number of categories, $\eta = (\eta_0, \eta_1, \dots, \eta_p)$ are the set of model parameters, $y^{(i)}$ is the ground truth value that corresponds to the label $d_i$, and $h_\eta(x^{(i)})$ is the predictive value.

The essence of network training is to update the parameters of the CNN model, which guarantees that the output of network conforms to the reality. As a result, with the optimization of Adam [41], the model parameters are updated along with the direction of gradient of $J(\eta)$. The update processes of parameters are as follows:

(1) The first and second order moments of gradient are calculated by (14) and (15), respectively:

$$m_t = \lambda_1 \cdot m_{t-1} + \frac{(1-\lambda_1)}{m}\sum_{i=1}^{m}\left(h_\eta\left(x^{(i)}\right) - y^{(i)}\right)x_j^{(i)} \tag{14}$$

$$n_t = \lambda_2 n_{t-1} + \frac{(1-\lambda_2)}{m^2}\left(\sum_{i=1}^{m}\left(h_\eta\left(x^{(i)}\right) - y^{(i)}\right)x_j^{(i)}\right)^2 \tag{15}$$

where $\lambda_1, \lambda_2 \in [0,1]]$ are the descent rates, and $t$ denotes the $t$th iteration.

(2) Then, the deviation of the first and second order moments are rectified by the correction coefficients:

$$\hat{m}_t = \frac{m_t}{1 - \lambda_1^t}, \hat{n}_t = \frac{n_t}{1 - \lambda_2^t} \tag{16}$$

(3) Finally, the learning rate of every parameter is updated based on the last gradient:

$$\eta_t = \eta_t - \frac{\tau\hat{m}_t}{\sqrt{n_t} + \delta} \cdot \frac{\partial J(\eta_t)}{\partial \eta_t} = \eta_t - \frac{\tau\hat{m}_t}{m(\sqrt{n_t} + \delta)}\sum_{i=1}^{m}\left(h_\eta\left(x^{(i)}\right) - y^{(i)}\right)x_j^{(i)} \tag{17}$$

From (14)–(17), it can be concluded that the learning rate of each parameter is updated with the dynamic change of the first and second order moments in the loss function. Meanwhile, an optimized gradient is obtained. Larger gradient would not bring larger step because the learning step of each iteration parameter is limited to a certain range, so the value of parameters is kept relatively stable.

The architecture of LatDisLanes is depicted in Figure 6, wherein the images are fed into the first convolution layer. After the image is fed into the network, the features are extracted by three convolution and two pooling layers. Moreover, the last dropout layer is accompanied by two fully connected layers. Finally, the softmax layer is introduced with the aim to obtain the probability distribution, where 322 results correspond to the lateral distance and one represents no lane in the image. Rectified linear units are used as activation function, and two dropout layers followed each pooling to overcome the over fitting.

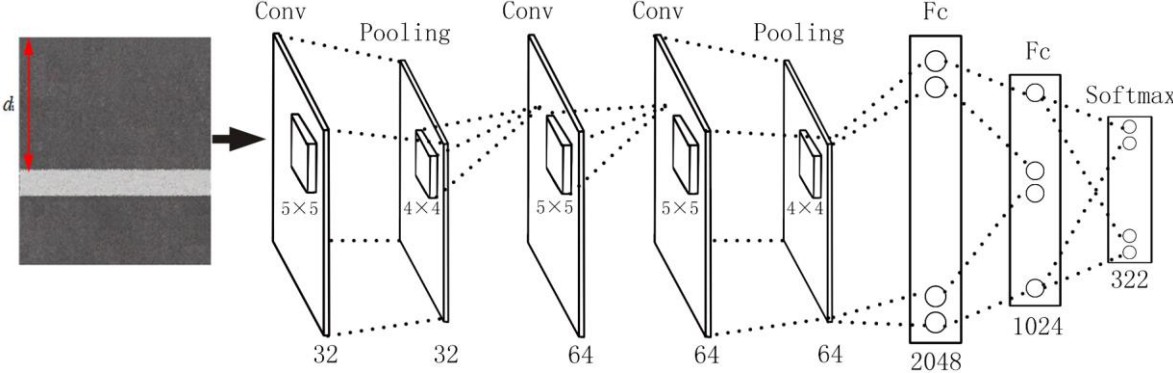

**Figure 6.** The LatDisLanes network architecture.

### 2.3. Dynamic Correction Model

In a smooth running condition, the deep learning model LatDisLanes has high recognition accuracy and a satisfactory recognition result. However, if the inclination angle changes drastically due to the fluctuation of road surface and centripetal force under high speed, larger recognition error will result. In this study, a dynamic correction model is developed by analyzing the influence of inclination angle on lateral distance recognition and the error is reduced in this way.

The camera position in the world coordinate system is depicted in Figure 7, wherein $\theta$, $\alpha$, and $\beta$ are inclination, yaw, and roll angle of the vehicle, respectively. Since this study focuses on the lateral distance between the lane markings and vehicle, the effect that is produced by $\alpha$ and $\beta$ is so little that it can be neglected. The $\theta$ is closely related to the lateral distance on *y*-axis.

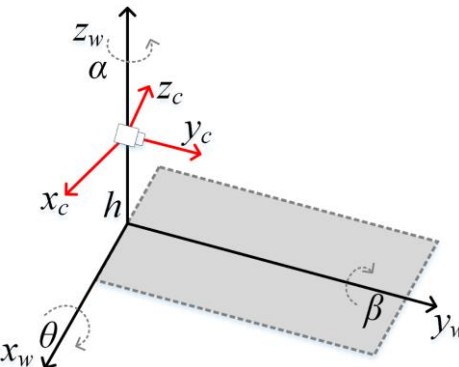

**Figure 7.** Three-dimensional (3D) world and camera coordinate systems, where $\theta$, $\alpha$, and $\beta$ are inclination, yaw and roll angle of the vehicle, respectively.

By studying the relationship among vehicle coordinates, camera coordinates, image coordinates, and pixel coordinates, this study ultimately obtains the variation of $\Delta y_w(\theta)$ in the world coordinate system induced by $\Delta\theta$. The process is as follows:

1. Transformation from world coordinate to camera coordinates

Let $P(x_w, y_w, z_w)$ be a point in the world coordinate system, then the corresponding point in camera coordinate system is $P'(x_c, y_c, z_c)$, and camera position in world coordinate is $x_0 = [0,0,h]$. The transformation from world coordinates to camera coordinates is obtained by:

$$x_c = R(x_w - x_0) \tag{18}$$

where $R = \begin{bmatrix} \cos\alpha & 0 & -\sin\alpha \\ -\sin\theta\sin\alpha & \cos\theta & -\cos\alpha\sin\theta \\ \cos\theta\sin\alpha & \sin\theta & \cos\theta\cos\alpha \end{bmatrix}$ is a rotation matrix.

2.    Transformation from camera coordinates to image coordinates

Suppose that point $P'(x_c, y_c, z_c)$ in the camera coordinate system corresponds to the point $Q(x, y)$ in image coordinate system, then if we let $f$ be a focal length of camera, according to the relationship of perspective projection, the transformation from camera coordinates to image coordinates is obtained by:

$$\begin{bmatrix} x \\ y \end{bmatrix} = \begin{bmatrix} f\frac{x_c}{z_c} \\ f\frac{y_c}{z_c} \end{bmatrix} \tag{19}$$

3.    Transformation from image coordinates to pixel coordinates

If the point $Q'(x, y)$ in pixel coordinate system corresponds to the point $Q(x, y)$ in the image, then the mapping between $Q$ and $Q'$ can be obtained by:

$$\begin{bmatrix} u \\ v \end{bmatrix} = \begin{bmatrix} s_u & s_1 & u_0 \\ 0 & s_v & v_0 \end{bmatrix} \begin{bmatrix} x \\ y \\ 1 \end{bmatrix} = \begin{bmatrix} f\frac{s_u x_c}{z_c} + u_0 \\ f\frac{y_c}{z_c} + v_0 \end{bmatrix} \tag{20}$$

where $(u_0, v_0)$ is the cardinal point, $s_u$ and $s_v$ are the pixel distances in the horizontal and vertical directions in images, and $s_1$ is the skew factor. The radial and tangential distortions are not taken into account in this study, thus $s_1$ is set to 1 [42].

As the goal of this study is to emphasize the lateral distance on the ground, $z_w$ is set to 0 and $y_w(\theta)$ is obtained by mapping from world coordinates to pixel coordinates:

$$y_w(\theta) = -\frac{h((u - u_0)\cos\alpha + \sin\alpha(f\cos\theta - v\sin\theta + v_0\sin\theta))}{f\sin\theta + v\cos\theta - v_0\cos\theta} \tag{21}$$

The lateral distance caused by a change of inclination angle is defined by:

$$\Delta y_w(\theta) = y_w(\theta + \Delta\theta) - y_w(\theta) \tag{22}$$

In fact, the inclination angle of vehicle is generally small and up to 5°, thus $\Delta y_w(\theta)$ can be approximated by:

$$\Delta y_w(\theta) = y'_w(\theta)\Delta\theta = \frac{h\sin\alpha(u_0 - u)(f\sin\theta + v\cos\theta - v_0\cos\theta)}{(f\sin\theta + v\cos\theta - v_0\cos v_0)^2}\Delta\theta \tag{23}$$

## 3. Experiment and Discussion

The experiments are conducted using the Ubuntu 14.04 operating system, i7-7600K CPU, 16 GB memory, and NVIDIA GTX1080 graphics card.

The Section consists of the following steps. Firstly, the improved Image Quilting and the proposed ground truth generation algorithm are used to generate the label images, and the similarity between the label images and the images in real scene are evaluated while using the Root Mean Square Error (RSME), Peak Signal to Noise Ratio (PSNR), and Structural Similarity (SSIM). Then, by using the generated label images to train this CNN model, a higher accuracy of lateral distance can be obtained under smooth conditions. Finally, the large error that is caused by the inclination angle was reduced by proposing a dynamic correction model.

### 3.1. Evaluation of the Improved Image Quilting

In this study, the Chinese expressway is used as the research object. According to the standard, the width of the lane on highway is 15 cm. Usually, the lateral distance between vehicle and the lane boundary is no more than 150 cm. Hence, the maximum distance is set to 150 cm. After camera

calibration, the length of 1 cm in the world coordinate system corresponds to 2.13 pixels in the image. As a consequence, the maximum distance $d_{max}$ and width of the lane in the image are 31.95 and 319.5 pixels, respectively. Therefore, the road size is set to $360 \times 360$ pixels. Moreover, the pixel distance between the lane strip and asphalt is $d_i$ ($d_i \in [0,320]$), which is used as a label in the process of image composition. There may be no lane in the natural traffic scene, and the label $d_i = 321$ is used to indicate that condition. The code of improved Image Quilting was programmed in MATLAB 2016 and MySQL 5.5. The images in real scene are captured by a camera on the rear-view mirror in the circular highway of Chongqing. In order to ensure the purity of the original images, they were collected in a sunny day without any shadow on the road. After the image acquisition was completed, we selected 1200 images for real scenes for compositing and similarity comparisons. According to the parameters that are presented in Table 1, 100 lane and asphalt images were synthesized.

**Table 1.** The parameters used for improved Image Quilting to synthesize images.

| Image Type | Texture Block Size (Pixel $\times$ Pixel) | Block Number | $\alpha$ | $\beta$ | Output Image Size (Pixel $\times$ Pixel) | Synthesis Time (s) |
|---|---|---|---|---|---|---|
| Lane | $8 \times 30$ | 2500 | 0.91 | 0.09 | $32 \times 360$ | 7.73 |
| Road | $30 \times 30$ | 2500 | 0.65 | 0.35 | $360 \times 360$ | 20.18 |

The average synthetic time of lane and asphalt by improved Image Quilting and the algorithms presented in [31,34] were shown in Figure 8a,b. When the value of $\lambda_0$ is small, then the probability of one-time successful selection of the required texture block in a random way is approximately 100%. The improved Image Quilting algorithm needs to calculate gradient and RGB information simultaneously, and the computational amount would be increased. Accordingly, the synthesis takes longer time. With the increase of $\lambda_0$, the probability of finding the required texture block using the algorithms presented in [31,34] was decreased and the synthetic time was increased drastically. As the required texture block can be searched in the database directly, the synthetic time in the improved Image Quilting is constant no matter how big $\lambda_0$ is.

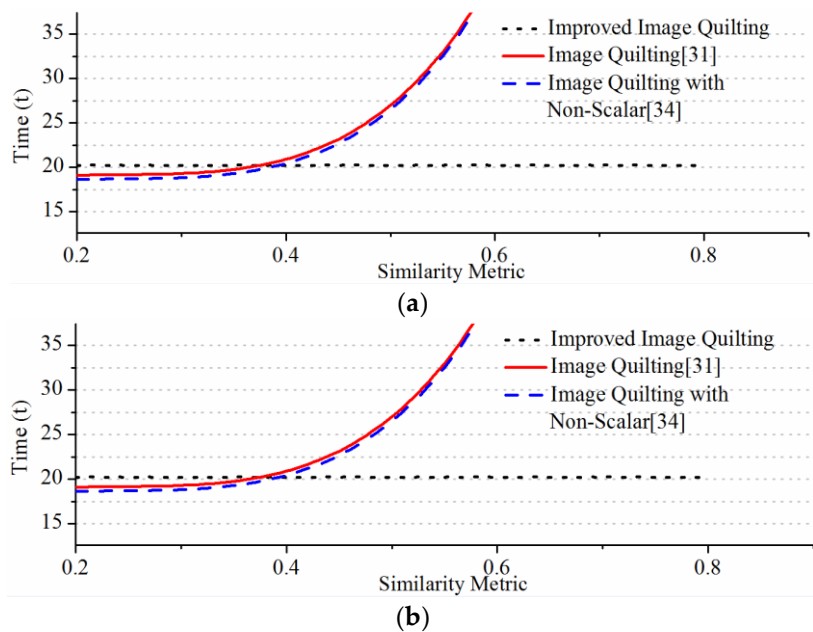

**Figure 8.** The mean synthesis time of lane in (**a**) and asphalt in (**b**) by the algorithm in [31,34], and by the improved Image Quilting in this study. The similarity metric along the horizontal axis is $\lambda_0$.

In fact, when a large number of images need to be synthesized, the speed can be improved greatly through parallel computing in MATLAB. An image with the size of $360 \times 360$ pixels can be synthesized

in 3 s. Therefore, a large number of high-quality images can be synthesized rapidly using the algorithm that is presented in this study.

To ensure the validity of the recognition model, the generated label images should be sufficiently similar with the real scene images. In this study, the similarity was evaluated by the full reference image quality assessment [43]. First, 100 images from four categories of road images were generated according to the damage degree [44]. The parameters are shown in Table 2.

**Table 2.** Road parameters.

| | Image Category | $N_l$ | $L_{th}$ | $H_{th}$ | $\gamma$ | $L_s$ | $c$ |
|---|---|---|---|---|---|---|---|
| I | No wear, no dirt, no shadows | 2 | 130 | 120 | 0.3 | 20 | 5 |
| II | Light wear, dirt, no shadows | 2 | 97 | 159 | 0.5 | 20 | 5 |
| III | Middle wear, dirt, shadows | 2 | 10 | 215 | 0.6 | 20 | 75 |
| IV | Severe wear, no dirt, shadows | 2 | 10 | 215 | 0.6 | 20 | 82 |

Four categories of images selected from the real scene were taken as a reference, and the quality of synthesized images was evaluated by RSME, PSNR, and SSIM, as the index of objective evaluation, PSNR, and RMSE are the most widely used [45]. Assume that the size of image was $H \times W$, then RMSE and PSNR were defined by:

$$RMSE = \left( \frac{1}{H \times W} \sum_{i=1}^{H} \sum_{j=1}^{W} (X(i,j) - Y(i,j))^2 \right)^{1/2} \tag{24}$$

$$PSNR = 20\lg \frac{(2^n - 1)}{RMSE} \tag{25}$$

where $X$ and $Y$ were the reference and generated image, respectively, and $n = 8$ was the number of sample bits of the image.

The average RMSE and PSNR of four categories of images were shown in Figure 9a,b, respectively. Since the size of images was $360 \times 360$ pixels and the average value of RMSE was less than 35, and PSNR was in the range [30,35], it meant that the values of RMSE and PSNR were within a reasonable range, indicating that the pixels of the generated images share some significant similarity with that of the images in real scene.

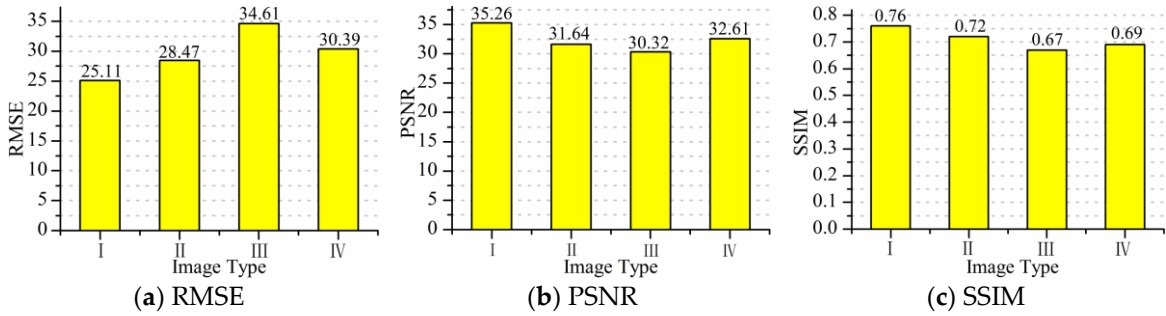

**(a)** RMSE        **(b)** PSNR        **(c)** SSIM

**Figure 9.** Evaluation results of label images. The three are the results of Root Mean Square Error (RMSE), Peak Signal to Noise Ratio (PSNR), and Structural Similarity (SSIM). Further, the I, II, III, and IV are four types of images.

Nevertheless, in (24) and (25), it can be seen that RMSE and PSNR can evaluate pixel similarity well regardless of structure information. However, the structure is not taken into account. SSIM is an objective evaluation index that is based on structure features, wherein the brightness, contrast, and structure similarity are taken into consideration. The result of SSIM is shown in Figure 9c. The values

were all greater than 0.65. Therefore, it can be concluded that the structure of label images was consistent with the structure of the images in the real scene.

From the experimental results that are mentioned above, we may draw the conclusion that the pixel and structure of the label images share a significant similarity with that of the images of the real scene, and the generated label images can meet the requirement of model training.

### 3.2. Evaluation of the LatDisLanes

Training the LatDisLanes model requires a lot of label images. 250,000 images were generated by the method that is presented in Section 2.1 and the distribution images are shown in Table 3.

**Table 3.** Images distribution.

| Dataset Name | Number |
|---|---|
| Training data | 175,000 |
| Validation data | 37,500 |
| Test data | 37,500 |

The LatDisLanes model that is proposed in this study was built with the Keras based on the Tensorflow. The initial values of the training parameters were as follows: maximal number of epochs was set to 40,000 (*max_epoches* = 40,000), learning rate was set to 0.001 (*learning_rate* = 0.001), batch size was set to 128 (*batchsize* = 128), and momentum was set to 0.9 (*momentum* = 0.9). The training of LatDisLanes was completed on the GTX1080 and it lasted for 6.5 h. The loss and accuracy are shown in Figure 10, wherein it can be seen that loss dropped to 0.16 and the accuracy rate reached 94.82% after 60 epochs.

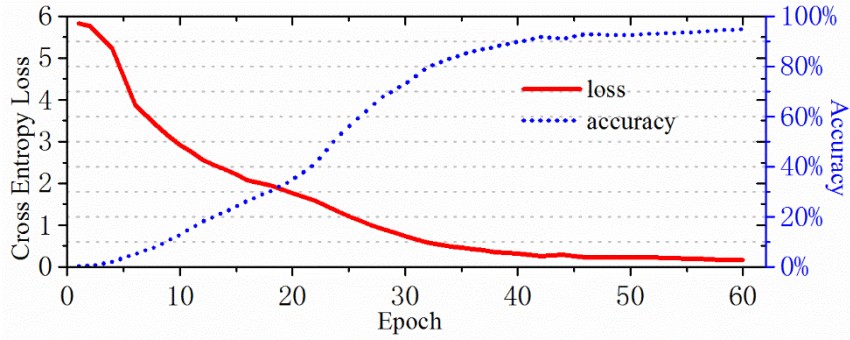

**Figure 10.** The process of loss and accuracy during model training. The red line is the loss and the blue dot line is the accuracy.

The pixel error was introduced as a standard metric. Namely, $E_k$ and $AE_k$ were used to compute the accuracy and absolute accuracy of the result in test dataset. The Iversion bracket $[P]$ was introduced, such that the value is equal to 1 if $P$ is true or 0 if $P$ is false [46]. The definitions of $E_k$ and $AE_k$ are:

$$E_k = \frac{1}{|V|}\sum_i (e_i - t_i)[|e_i - t_i| \le k], k \in \{0, \ldots, 5\}, \tag{26}$$

$$AE_k = \frac{1}{|V|}\sum_i |e_i - t_i|[|e_i - t_i| \le k], k \in \{0, \ldots, 5\}, \tag{27}$$

The value of $AE_k$ is shown in Figure 11a, wherein the accuracy of recognition of distance that less than 1 and 5 pixels as compared with the ground truth, which were 94.78% and 99.94%, respectively. Since the pixel length in image corresponded to the length of 0.46 cm in the world coordinate system, the accuracy was 94.78% if the allowable error was 0.46 cm and 99.94% if the allowable error was 2.3 cm.

Furthermore, the distribution of $E_k$ was shown in Figure 11b, wherein the accuracy was decreased along with the distance on both sides. It can be seen that the accuracy of the lateral distance recognition model based on the CNN that was designed in this study reached a high level, which shows that the synthesized image can satisfy the needs of model training.

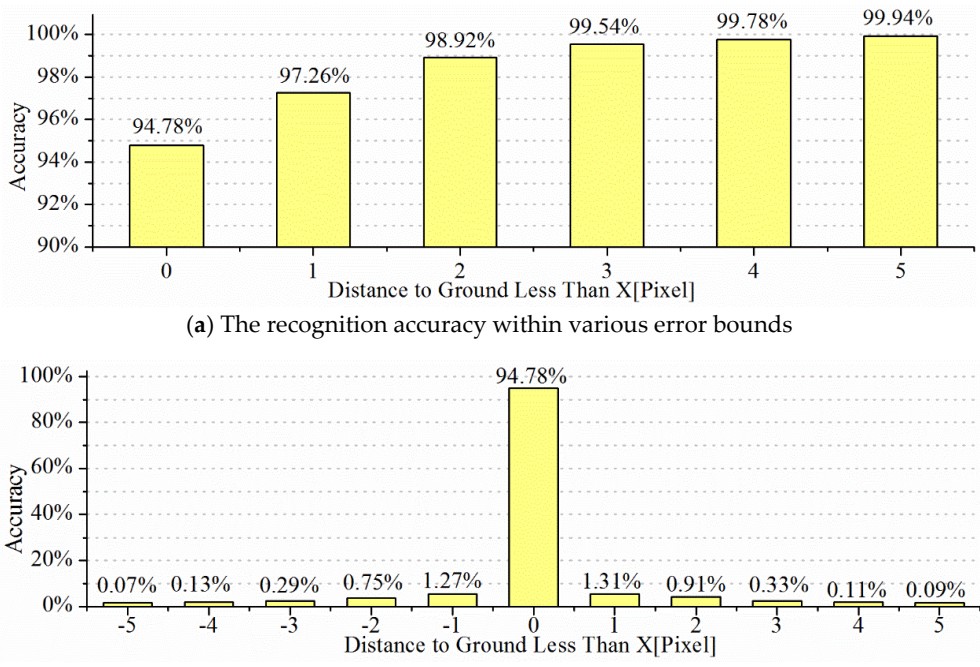

(**a**) The recognition accuracy within various error bounds

(**b**) The deviation between ground truth and network output

**Figure 11.** The data set classified within certain error bounds $|e_i - t_i| < k$ with $k \in \{0, \dots, 5\}$. The image row containing the lane marking can be predicted in 97.26% within a precision of 1 pixel, and in 99.94% of the cases with 5-pixel precision in (**a**). (**b**) shows the evaluation result of $e_i - t_i$ in test images.

## 3.3. Experiment of the Corrected Model

During driving, due to the unbalanced load, turning, road bump, and other factors, a vehicle does not move in a stable condition. In this study, the inclination angle has great influence on the lateral distance recognition. By constructing a vehicle dynamics correction model, the relationship between the inclination angle and the recognition distance error that is shown in Equation (23) is obtained. Therefore, only by measuring the inclination angle of the vehicle can the recognition error that is caused by the inclination angle in real time be corrected.

The experiment was carried out in Chongqing Intelligent Vehicle Integrated System Test Area (i-VISTA) and part of time series were selected to analyze the method efficiency. The inclination angle of the vehicle was obtained by a gyroscope and the noise that was subjected to normal distribution was filtered out by Gaussian [47,48]. It is defined that the vehicle's inclination angle is 0 at equilibrium, and the positive was to the left and the negative was to the right. After the completion of the test, the sequence of inclination angle and recognition result is obtained. To evaluate the validity of the dynamic correction model, a sequence in the test process was selected for analysis and the ground truth was manually labelled by the time-slice method. As shown in Figure 12, the range of the inclination angle is generally small, fluctuating in the interval $[-4,4]$.

The result of recognition by LatDisLanes with a high recognition accuracy was consistent with the ground truth, and the average errors on the left and right sides were 0.871 and 0.865 cm, respectively. As shown in Figure 13a,c, in the vicinity of the 6000th, 8600th, and 9400th frames, the inclination angle was greater than $3°$, which caused a big error. After the error was rectified by the dynamic correction model, the result was closer to the ground truth, as shown in Figure 13b,d. Meanwhile, the average errors on the left and right sides decreased to 0.791 and 0.765 cm.

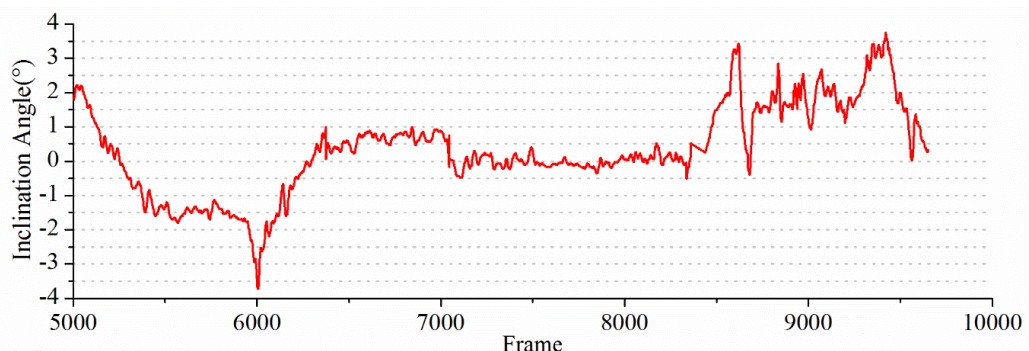

**Figure 12.** Inclination angle of vehicle.

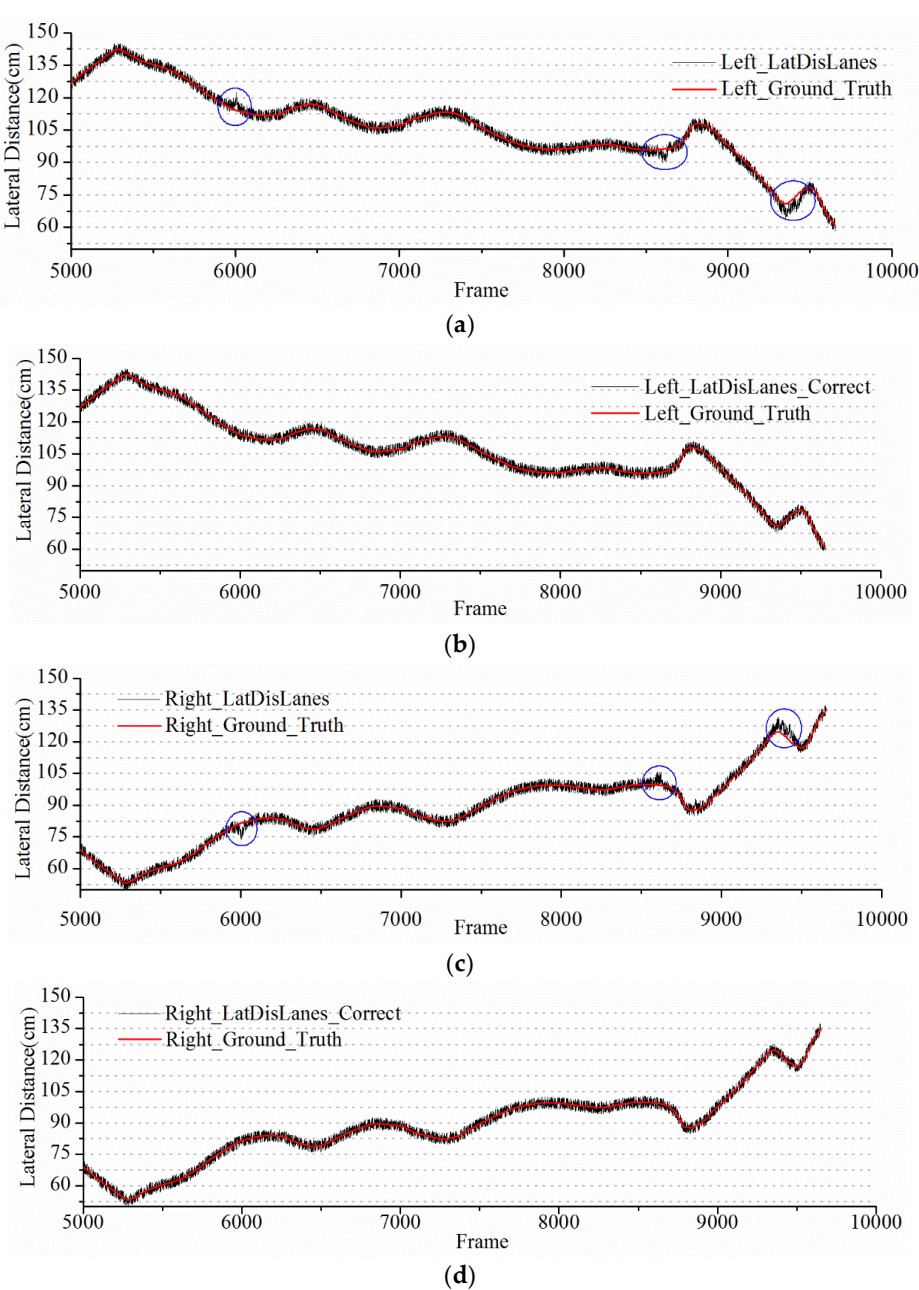

**Figure 13.** The red line is the ground truth. The black in (**a**,**b**) is the output of DisLanes, wherein the black line away from the red that means larger error in the circle. The black line fluctuates in a small range compared to the red after the correct model in (**c**,**d**).

However, the model was based on machine vision and was highly sensitive to light brightness. In the nighttime situation, since the illuminating light was located in front of the vehicle, the side light was extremely weak. It was difficult for the model to detect accurately without supplementary light source. In order to test the performance of the system at night, we have added the test in the circular highway at night with the conditions of streetlights, no streetlights, and a lateral supplementary light. The average error was still as the criteria to evaluate the performance of the model. The test results are shown in Table 4. Obviously, the detection result was better when there is an auxiliary light source, which was basically consistent with the result in the daytime. However, when there was only the street light, the average error fell to 4.296 cm. If there is no street light, then the average error drops to 53.472 cm, which made the detection meaningless. Therefore, in order to ensure the accuracy of the model, a lateral supplementary light source is necessary.

**Table 4.** Detection results of models at night.

| Light Condition | Average Error (cm) |
| --- | --- |
| Lateral supplementary light | 0.825 |
| Streetlights | 4.296 |
| No streetlights | 53.472 |

## 4. Conclusions

Currently, it is of great importance to accurately locate the lateral position of a vehicle in the lane in order to facilitate the road test of these ADASs, such as LDW and LKAS. On the basis of the above, this study draws the following conclusions:

1. An algorithm for image synthesis is proposed based on the Image Quilting by improving the metric of texture block matching and method of block selection, which can rapidly synthesize numerous high-quality lane and asphalt images.
2. A label automatic annotation algorithm is presented, wherein the distance between the edge of lane and asphalt was set as the label, and factors of light, road wear, and dirt were taken into account. The experiment shows that the label images obtained had a high quality and they could be used for the model training.
3. By constructing a deep CNN, LatDisLanes, the model can recognize the lateral distance with a sub-centimeter precision. Furthermore, a dynamic correction model is proposed to reduce the recognition error that is caused by the inclination angle of vehicle. The experiment was conducted in i-VISTA and the result showed that inclination angle greater than $3°$ would cause a bigger error, which can be reduced by the proposed correction model.

In this study, a lateral lane distance prediction framework is constructed systematically, and the output reaches a sub-centimeter precision. However, the lane is determined by scanning the whole image with the deep neural network, the large computational amount slows down the recognition. With the rapid development of object detection algorithms, such as YOLO and SSD etc., we will further study the high-precision and high-speed lane detection algorithm using these object detection algorithms.

**Author Contributions:** X.Z. and W.Y. conceived and designed the method and performed experiments; X.T. reviewed and edited this manuscript; Z.H. carried out part of the experimental verification.

**Funding:** This research was funded by Chongqing Natural Science Foundation, grant number No. cstc2018jcyjAX0468, National Natural Science Foundation of China, grant number No. 51705044, and Graduate Scientific Research and Innovation Foundation of Chongqing, China, grant number No. CXB226.

**Conflicts of Interest:** The authors declare no conflict of interest.

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
