# Peer review of "Estimation of the Lateral Distance between Vehicle and Lanes Using Convolutional Neural Network and Vehicle Dynamics"

_applsci, doi:10.3390/app8122508_

Round 1
Reviewer 1 Report
The paper deals with a recognition model to estimate the distance between vehicle and lane boundaries by training a convolutional neural network model. The distance is obtained by using two cameras mounted at the side mirrors and it can be estimated directly since the model does not need any data pre-processing and post-processing.
In fact, an image synthesis algorithm allows to collect a small number of images in a real scene and produces a large number of images as needed; then, a label automatic annotation algorithm provides the label images for training the neural network; finally, the algorithm calculates the lateral distance.
The model can be proposed to test and evaluate the performance of some applications in ADASs, such as LKAS, LDW; it can also be used in the design of these kind of applications.
The results show that the model achieve a very high accuracy of recognition but it is disrupted by the inclination angle. This last factor can be corrected by using a correction model based on vehicle dynamics.
In general, the paper is clear and methodologically correct; the review of literature is wide and proper. The title and abstract are appropriate; the subject fits with the journal’s aims and scope.
The research is quite innovative and interesting for readers, especially researchers and technicians that are developing new systems and algorithms in ADASs.
Only a few suggestion will be proposed in the following, to further improve the paper:
- in some cases, the word “lane” is used with the meaning of lane boundary or lane strip; this circumstance can create possible confusion, especially when dimensions or positions are provided with reference to this element;
- Authors affirm that in the study the algorithm only needs a small number of images in real scenes and, with this basis, it can produce a large number of synthesized images; however, the number of real images actually used in the proposed calibration and validation case study seems not clear enough. In line 363, in fact, there is this sentence: “…100 lane and asphalt images were synthesized”; so, this number seems to refer to sinthesized and not to real-scene images. It is important to better explain how many real images have been used and where they have been acquired (in which condition: weather, local environment, shadows, asphalt characteristics and conditions), as well as how many synthetic images have been produced.
- In any case, the size of image sample seems too small to validate the proposed model, especially if compared with other similar studies (just as an example, the Authors affirm, in the introduction, that a previous study based the results on 80,000 real scene images and 40,000 synthetic images). Thus, an importan advise is to increase the number of images in the proposed analysis.
Author Response
Dear reviewer:
We are sincerely grateful for the insightful and constructive comments concerning our manuscript entitled "Estimation of the lateral distance between vehicle and lanes using convolutional neural network and vehicle dynamics". Those comments are all valuable and very helpful for revising and improving our paper, as well as the important guiding significance to our researches. We have studied comments carefully and have made correction which we hope meet with approval. In the attachment, we carefully present our response to the comments.
Kind regards,
Wei Yang

Reviewer 2 Report
This paper proposes a neural network-based recognition model to estimate the distance between the vehicle and the lane boundaries. This paper also proposes some additional techniques for image synthesis, automatic label annotation and correction model to reduce errors etc for precise estimation.
Overall, the paper is well written, and it is easy to read. But there are few things that need to be addressed. I suggest the manuscript to be revised at least in the following ways.
The authors used original images and then generated synthesis images as inputs for CNN. Few road parameters are considered like wear, dirt and shadow but no other lighting conditions like day/night. In this case, the performance highly depends on whether the image capturing conditions between the training and testing dataset are similar or not. This makes the algorithm less robust to the variations of different datasets. Can the images captured during the night (with a flashlight) be detectable by the proposed method? There should be some experiment in paper where test images captured at night are used to measure the performance of the system.
An experiment to verify the performance of proposed techniques on any other available dataset like Kitti or cityscape can significantly improve the quality of the paper.
Author Response

(The authors gave the same response as above.)

Reviewer 3 Report
This topic has been studied in depth in the past using other tools. Authors should explain the reason thay think their method could improve results obteined with other methods. The method seems complex considering its purpose.
The abstract should introdce the tipoc and perhaps so many details of what has been done are not necessary.
The first sentence is not copeltely correct. There arether tools apart from computer visin and big data. And big data is not essential for many ADAS.
Please, revise that the references in thext are correctly cited. I think that some refeences cited in the text do not correspond with the reference included at the end. Or, at least, it is not clear why they are cited.
Why the variable frame is used in figures 11 and 12 and start in 5000? Perhaps, time could be better...
Some figure legends include discussion of results and other not. Pleaase, make them homogeneus.
The paper is difficult to read. I suggest some rewritting to make it more comprehensible.
Author Response

(The authors gave the same response as above.)

Round 2
Reviewer 1 Report
The Authors have correctly addressed the most of comments and their answers clarified some of the main problems that I recognized in the previous version.
The paper now presents a research that seems quite similar to ones previously published by other researchers, but methods and algorithms are original.
To better explore this kind of methods, in my opinion, a wider sample of real scenes, including different weather and lighting conditions or shadows, colors and spots on the carriageway, will be needed to be collected in future researches.
Author Response
Dear reviewer:
We are sincerely grateful for the insightful and constructive comments concerning our manuscript entitled “Estimation of the lateral distance between vehicle and lanes using convolutional neural network and vehicle dynamics”. Those comments are very valuable and helpful for us to revise our manuscript, as well as the important guiding significance to our researches. We have studied comments carefully and have made correction which we hope meet with approval. In the following, we carefully present our response to the comments, and the file was added in the attachment.

Reviewer 2 Report
The authors answered each question properly and made reasonable modifications accordingly. I think the current version has been improved significantly and will be ready to publish after a minor spell check.
However, i would suggest to include diverse dataset for proper comparison for future ongoing work.
Author Response

(The authors gave the same response as above.)

Reviewer 3 Report
All my previous comments have been adressed. Just one of them is still unclear. When I pointed out that the paper was difficult to be read it was not exclusevely because of English but the structure. I suggest some clarification on the structure and how it is presented.
Author Response

(The authors gave the same response as above.)
